# Exploring Vector Spaces for Semantic Relations

## Abstract

Word embeddings are used with success for a variety of tasks involving lexical semantic similarities between individual words. Using unsupervised methods and just cosine similarity, encouraging results were obtained for analogical similarities. In this paper, we explore the potential of pre-trained word embeddings to identify generic types of semantic relations in an unsupervised experiment. We propose a new relational similarity measure based on the combination of word2vec's CBOW input and output vectors which outperforms concurrent vector representations, when used for unsupervised clustering on SemEval 2010 Relation Classification data.

## 1 Introduction

### 1.1 Vector space semantics

Vector space word representations or *word embeddings*, both 'count' models (Turney and Pantel, 2010) and learned vectors (Mikolov et al., 2013a; Pennington et al., 2014), were proven useful for a variety of semantic tasks (Mikolov et al., 2013b; Baroni et al., 2014). Word vectors are used with success because they capture a notion of semantics directly extracted from corpora. Distributional representations allow to compute a functional or topical semantic similarity between two words or, more recently, bigger text units (Le and Mikolov, 2014). The closer two entities are in the vector space (quantified usually, but not necessarily in terms of cosine similarity), the more similar they are semantically. This similarity can be exploited for lexical substitution, synonym detection, subcategorization learning etc. Recent studies suggest that neural word embeddings show higher performance than count models (Baroni et al., 2014; Krebs and Paperno, 2016) for most semantic tasks, although Levy et al. (2015a) argue that this is only due to some specific hyperparameters that can be adapted to count vectors. In what follows, we will concentrate on exploring whether and how pre-trained, general-purpose word embeddings encode relational similarities.

### 1.2 Relational analogies as vector offsets

Relation extraction and classification deal with identifying the semantic relation linking two entities or concepts based on different kinds of information, such as their respective contexts, their co-occurrences in a corpus and their position in an ontology or other kind of semantic hierarchy. Whether the vector spaces of pre-trained word embeddings are appropriate for discovering or identifying *relational* similarities remains to be seen. Mikolov et al. (2013b) claimed that the embeddings created by a recursive neural network indeed encode a specific kind of relational similarities, i.e. *analogies* between pairs of words. He found that by using simple vector arithmetic, analogy questions in the form of "$a_1$ is to $a_2$ as $b_1$ is to $b_2$" (*man ~king :: woman ~queen*) could be solved. Relationships are assumed to be present as vector offsets, so that in the embedding space, all pairs of words sharing a particular relation are related by the same constant offset. Vector arithmetics give us the vector which fills the analogy, and we can search for the word $b_2$ whose embedding vector has the greatest simi-

larity to it:

$$argmax_{b_2} = sim(b_2, (b_1 - a_1 + a_2)) \quad (1)$$

Levy et al. (2015a) suggested that instead of a vector offset method, this calculation can also be considered as a combination of similarities. Using cosine similarity for $sim$, equation 1 can be written as a combination of similarities (Levy et al., 2015a) as

$$argmax_{b_2} = sim(b_2, b_1) - sim(b_2, a_1) + \\ + sim(b_2, a_2) \quad (2)$$

Analogy pairs, however, are a special case of relational similarity because not only $a_1$ ($man$) relates to $a_2$ ($king$) the same way that $b_1$ ($woman$) relates to $b_2$ ($queen$); the relation between $a_1$ ($man$) and $b_1$ ($woman$) is also parallel to the relation between $a_2$ ($king$) and $b_2$ ($queen$.) When it comes to different types of semantic relations, their instances may or may not be analogical: e.g., while we can say that a ($claw$) is a component of an ($owl$) just like ($walls$) are a component of a ($hospital$), no meaningful relational similarity is shared between the pairs ($claw$) $\rightarrow$ ($walls$) and ($owl$) $\rightarrow$ ($hospital$) (see Figure 1).

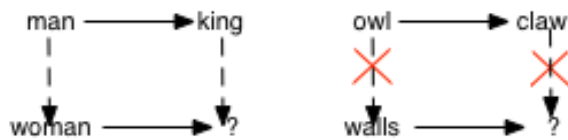

Figure 1: Analogical vs non-analogical semantic relation

### 1.3 Criticism of the vector offset method

As precise as neural word embeddings combined with cosine similarity may be for calculating semantic proximity between individual words, recent results seem to suggest that their value in identifying relational analogies using vector arithmetics is limited. In fact, a big part of their merits is likely to come from the precise calculation of individual similarities instead of relational similarities. Hence, they can be approximated using relation-independent baselines. Linzen (2016) remarks that currently used analogy tasks evaluate not only the consistency of the offsets $a_1 - a_2$ and $b_1 - b_2$, but also the neighborhood structure of the words in the vector space. Concretely, "if $a_1$ and $a_2$ are very similar to each other (...) the nearest word to $b_2$ may simply be the nearest neighbor of $b_1$ (...) regardless of offset consistency" (Linzen, 2016). Moreover, some of the success obtained by the vector offset method on analogies can also be obtained by baselines that ignore $a_2$, or even both $a_1$ and $a_2$.

Levy et al. (2015b) point out similar limitations of this line of work. Different word embedding combinations in supervised learning of taxonomical relations do not seem to learn the relations themselves, but individual properties of words. They tested previously suggested vector compositions for supervised learning of inference relations: concatenation, difference, comparing only the first or only the second element of the pairs. The study concludes that the classifiers only learn individual properties (e.g. a "category" type word is a good hyponym candidate), but not semantic relations between words.

These studies suggest that the semantic information obtained from word embeddings is correct for identifying similar or related units, but is already self-contained and difficult to enrich in order to retrieve more specific semantic contents such as relational similarities or specific relations.

In this paper, we aim to challenge this conclusion within a large scale semantic relation classification experiment, and show that it is possible to achieve improvements compared to baselines and current methods. We apply known vector composition methods, and propose a new one, to unsupervised large-scale clustering of entity pairs categorized according to their semantic relation. While large scale semantic relation classification is a very difficult task and the state of the art does not perform close to a human level, we expect that the experiment provides information to compare the semantic potential of different vector/similarity combinations.

## 2 Semantic Relations in Vector Spaces

### 2.1 Related work

Relation classification includes the task of finding the instances of the semantic relations, i.e. the entity tuples, and categorizing their relation according to an existing typology. In an unsupervised framework, relation types are inferred directly from the data. Supervised systems rely on a list of pre-defined relations and categorized examples, as described in the shared tasks of MUC, ACE or SemEval campaigns (Hobbs and Riloff, 2010; Jurgens et al., 2012; Hendrickx et al., 2010). Competing systems extract different kinds of features eventually combined with external knowledge sources, and build classifiers to categorize new relationship mentions (Zhou et al., 2005). A commonly used method, initiated by Turney (2005; 2006), is to represent entity couples by a pair-pattern matrix and calculate similarities over the distribution of the couples. Another way of constructing a distributional vector space to represent quantifiable context features for relation extraction is to combine the vectors of the two entities. Different combination methods were proposed to represent compositional meaning (Mitchell and Lapata, 2010; Baroni and Zamparelli, 2010; Baroni et al., 2012). Popular methods include addition (Mitchell and Lapata, 2010), concatenating the two vectors (Baroni et al., 2012) or taking their difference (Weeds et al., 2014). As of now, these vector combinations had two types of applications in semantic relation classification. The first one aims to find specific types of semantic or functional analogies (Herdağdelen and Baroni, 2009; Makrai et al., 2013; Levy et al., 2015a). The second one tries to infer taxonomical relations in supervised experiments (Weeds et al., 2014; Turney and Mohammad, 2014). Recently, Turney (2012) suggested a dual distributional feature space for supervised classification. Herdağdelen and Baroni (2009) combine individual entity vectors with co-occurrence contexts in their vector space. These experiments either aim to identify very specific relation types (typically taxonomical relations) with a mixture of features and a supervised classifier, or target analogy pairs: a task in which, as we have seen, relation-unaware

baselines approximate relation-aware representations.

### 2.2 Task definition

Whether we use the vector offset method or any pairwise similarity combination, finding the missing word in an analogy depends on two factors:

1. Vector quality (do semantically close elements have a higher cosine similarity?)

2. Density and structure of the vector space.

If we adapt 1) above to the more generic relational similarity task, the question can be formulated as follows:

3. How much information about the semantic relation is actually in the text and how fit is the vector combination method to encode this information?

In accordance with Linzen (2016) and Levy et al. (2015b), we also think that analogy test sets are not optimal to answer this question. We propose to study relational similarity using a more generic and large scale relation classification task (Hendrickx et al., 2010), and clustering pairs according to semantic relations, instead of finding the one missing word in an analogy. This way, we rely less on the neighborhood structure and more on actual "linguistic regularities".

We evaluate different vector combination methods, and propose a new one, for calculating relational similarities. The evaluation concentrates on the aspects above. We test whether cosine similarity over these vector spaces is adapted for discovering groups and classifying individual instances. We report clustering results and compare the vector combinations by their performance.

### 2.3 Lexical vs contextual relations

The semantic relation classification task, supervised or not, is a difficult one with a strong upper bound: not every piece of relational information is explicit in the text. Some relations are more lexical by nature: relations such as *dog* is an *animal*; a *teacher* works at a *school*; a *car* is kept at a *parking lot*, can be interpreted independently of the context.

On the other hand, many relation instances are contextual. Contextual relations (e.g. "the *accident* was caused by the *woman*") tend to be expressed explicitly, but rarely, in a corpus. They can be handled by pattern-based approaches rather than distributional representation combinations. We expect to be able to identify prototypical lexical relation instances with vector combination methods. In the scope of the current experiment, we do not intend to combine contextual information with lexical semantic representations, since our primary goal is to argue that vector combinations may encode lexical relational similarities in themselves. If a representation is more capable than others to group together word pairs according to relational similarities, this potential can further be exploited in unsupervised as well as in supervised experiments. On the other hand, well-known outliers (contextual relations and less typical examples) will require a complementary approach.

This task is difficult and requires a change of perspective: when we look for missing elements in an analogy we know the word exists and we presume to know where it will be in the vector space, while in unsupervised clustering, our aim is to infer a global structure from the data.

### 2.4 Semantic relation data

The SemEval 2010 Task 8 data we used (Hendrickx et al., 2010) contains examples of relation instances for 9 relations with sufficiently broad coverage to be of general and practical interest (Table 1).

There is no overlap between classes, but there are two groups of strongly related relations to assess models' ability to make fine-grained distinctions (CONTENT-CONTAINER, COMPONENT-WHOLE, MEMBER-COLLECTION and ENTITY-ORIGIN, ENTITY-DESTINATION). This data set is very challenging, not only because of the fine semantic distinctions, but also because semantic relations were annotated in context and contain many less typical relation instances. In the current experiment, the goal we set for ourselves is to explore models' abilities to capture the structure of the data, rather then in achieving a classification precision close to that of humans.

We used 6637 pairs of single word instances from the training data. Contexts in the training data were discarded. Class bias is present: the most frequent relation has 979 instances, the least frequent has 486.

## 3 Vector combination methods

If $a_1, a_2, b_1, b_2$ are entities (nouns or nominal compositions) from a corpus, each of them assigned a pre-trained word embedding, we would like to classify entity couples $a = (a_1, a_2)$ and $b = (b_1, b_2)$ according to their semantic relation. This means that we are looking for an efficient combination of $a_1, a_2$ and $b_1, b_2$ vectors that encode their relational attributes. We aim to find effective methods to calculate a relational similarity $sim(a, b)$ by combining entity vectors $a_1, a_2$ and $b_1, b_2$.

**Pairwise similarities** build on the idea that if $a_1$ is semantically similar to $b_1$ and $a_2$ is similar to $b_2$, the relation between $a_1$ and $a_2$ is similar to the relation between $b_1$ and $b_2$. The recall of this approach is expected to be limited: the same relation can hold between different types of entities.

**Analogical similarities** presume that $b_1 - b_2$ shares the direction with $a_1 - a_2$, ignoring the pairwise similarities. We adapt this measure, while aware that analogy pairs are a specific case of relational similarity in that analogies work both ways (*man ~king* :: *woman ~queen* and also *man ~woman* :: *king ~queen*).

**IN-OUT similarities**: a new combination that builds on the integration of second order similarities.

**Only** $a_1$ : In this baseline solution, the similarity between two pairs is calculated as the similarity between the first entity of each pair, the other pair being ignored.

$$sim(a, b) = sim(a_1, b_1) \qquad (3)$$

### 3.1 Pairwise similarities

Different combinations proposed in the literature were compared.

- **concatenative** : one vector for each entity couple is defined as the concatenation of the vectors of the two entities.

$$sim(a, b) = sim((a_1 \oplus a_2), (b_1 \oplus b_2)) \quad (4)$$

| Relation | Instances in training data | Typical examples | Atypical examples |
|---|---|---|---|
| Cause-Effect | 979 | $suicide - death, injury - discomfort$ | $women - accident$ |
| Component-Whole | 978 | $claw - owl, walls - hospital$ | $image - photos$ |
| Entity-Destination | 789 | $solvent - flask, hay - barn$ | $chair - corporation$ |
| Product-Producer | 775 | $industry - models, artist - design$ | $officer - oath$ |
| Entity-Origin | 762 | $relics - culture, plane - runway$ | $error - definition$ |
| Member-Collection | 729 | $stable - hounds, ensemble - ladies$ | $mission - monkeys$ |
| Message-Topic | 622 | $pages - scene, speech - measures$ | $exhibition - glamour$ |
| Instrument-Agency | 517 | $user - console, eye - telescope$ | $companies - governments$ |
| Content-Container | 486 | $document - folder, pictures - box$ | $message - paper$ |

Table 1: Semantic Relation Classification data

- **pairwise addition** Pairwise similarities between respective entities are added up. If we use cosine similarity, this is only slightly different from the concatenative method. Vector addition proved to work well as a compositional representation (Mitchell and Lapata, 2010), despite the fact that word order is ignored.

$$sim(a, b) = sim(a_1, b_1) + sim(a_2, b_2) \quad (5)$$

A potential problem with this addition objective is that different properties of words are expressed on a different scale and, as a consequence, terms sharing these properties have a higher cosine similarity than terms that are similar with respect to a flatter property. It can be overcome by using multiplication instead of addition (Levy and Goldberg, 2014):

- **pairwise multiplication**

$$sim(a, b) = sim(a_1, b_1) \times sim(a_2, b_2) \quad (6)$$

### 3.2 Analogies

This is an adaptation of the measure proposed for $queen = king - man + woman$ (Mikolov et al., 2013b). Vector arithmetics give us the vector which fills the analogy, and we can search for the word $b_2$ whose embedding vector has the greatest similarity to it:

$$argmax_{b_2} = sim(b_2, (b_1 - a_1 + a_2)) \quad (7)$$

which, using cosine similarity for $sim$, can be written as a combination of similarities (Levy et al., 2015a) as

$$argmax_{b_2} = sim(b_2, b_1) - sim(b_2, a_1) + sim(b_2, a_2) \quad (8)$$

Mikolov (2013b) notes that this measure is qualitatively similar to the relational similarity model in (Turney, 2012), which predicts similarity between members of the word pairs $(x_b, x_d), (x_c, x_d)$ and dissimilarity for $(x_a, x_d)$. In the current context, we do not look for the missing $b_2$ which maximizes the equation. Instead, we have different couples $a$ and $b$, and we aim to calculate $sim(a, b)$ to quantify how much the analogy $queen - woman = king - man$ holds.

- **difference** Focuses on the similarity of $b_1$, $b_2$ and $a_1$, $a_2$, but does not take into account the pairwise distances between the individual entity vectors.

$$sim(a, b) = sim((a_1 - a_2), (b_1 - b_2)) \quad (9)$$

Levy et al. (2015a) propose a multiplicative version of the analogy formula. We tried to adapt it; however, this measure is not symmetrical (conceived to find $b_2$ which maximizes the form) and the adaptation gave bad results.

### 3.3 IN-OUT similarities

This metric is a combination of first order and second order similarities between the two entity pairs, adapted to relational similarity : $a$ and $b$ are similar if $a_1$ is similar to $b_1$ and also similar to the *contexts of* $b_2$, the opposite entity in $b$. In the current experiment, second order similarities are estimated using both input and output vectors generated by word2vec's CBOW model. In this model, the IN vectors of words get closer to the OUT vectors of other words that they co-occur with. Words with a high input-output similarity tend to appear in the context of each other. This similarity combination was shown to improve information retrieval (Nalisnick et al., 2016).

Also, Pennington et al. (2014) use second-order similarity to improve similarity calculation *between words*. Their proposed formula combines first and second order similarity, normalized by the reflective second order similarity of the words with themselves. This is based on the observation that "words are similar if they tend to appear in similar contexts, or if they tend to appear in the contexts of each other (and preferably both)." (Note that they use 'first order' for word-context (IN-OUT) similarity and 'second order' for word-word (IN-IN) similarity.)

$$sim(x,y) = \frac{sim_2(x,y) + sim_1(x,y)}{2\sqrt{sim_1(x,x)+1}\sqrt{sim_1(y,y)+1}} \quad (10)$$

In our experiment, second order similarities are used in a different way and with a different purpose. Second-order similarities are calculated between opposite elements of the entity couples. We combine those similarities by taking the in-in similarity between $a_1$ and $b_1$, and the in-out similarities between $a_1$ and $b_2$, and between $a_2$ and $b_1$.

- **additive in-out**

$$sim(a,b) = sim(a_1,b_1) + sim(a_2,b_2) \\ + sim_2(a_1,b_2) + sim_2(a_2,b_1) \quad (11)$$

where $sim_2$ designates the second order similarity and is calculated as follows:

$$sim_2(x_1,y_2) = sim(x_1^{in}, y_2^{out}) + sim(x_1^{out}, y_2^{in}) \quad (12)$$

- **multiplicative in-out**: The same as above, but addition is replaced by multiplication in $sim$ and $sim_2$.

$$sim(a,b) = sim(a_1,b_1) * sim(a_2,b_2) * sim_2 \\ (a_1,b_2) * sim_2(a_2,b_1)$$

where
$$sim_2(x_1,y_2) = sim(x_1^{in}, y_2^{out}) * sim(x_1^{out}, y_2^{in})$$

## 4 Clustering Experiments

We trained a word2vec CBOW model (Mikolov et al., 2013a) with negative sampling and a window size of 10 words on the ukWaC corpus (Baroni et al., 2009), and extracted both input

and output vectors of size = 400 to build the vector combinations above. An adjacency matrix was constructed for each vector/similarity combination using cosine similarity. Clustering was implemented with Cluto's (Zhao et al., 2005) clustering function which takes the adjacency matrix as input. We used a hierarchical agglomerative clustering with the unweighted average distance (UPGMA) criterion function[1].

### 4.1 Evaluation as classification

At first, we ran the clustering with 9 clusters (the number of classes in the standard) and tried to make one-to-one correspondences between the standard and the output. Every cluster is mapped to the standard class that shares the more elements with. We then calculate precision and recall for each standard class (zero if the class doesn't show up as a majority class in any cluster). Average class-based precision and recall is reported, as well as the number of classes in the standard that could be assigned. These scores were published for the SemEval task participants, but ours are not comparable because we only consider one cluster for each class, and because we did the clustering on the training data.

| INPUT | classes found | P | R | F |
|---|---|---|---|---|
| $a_1$(base) | 5 | 0.1700 | 0.2086 | 0.1873 |
| add | 6 | 0.1918 | 0.1973 | 0.1945 |
| conc | 6 | 0.2031 | 0.2115 | 0.2072 |
| in-out.add | 8 | 0.2635 | **0.2192** | **0.2393** |
| mult | 7 | 0.1824 | 0.1493 | 0.1642 |
| in-out.mult | 5 | 0.1102 | 0.1232 | 0.1163 |
| diff | 5 | **0.3762** | 0.0918 | 0.1476 |

Table 2: Class-based results for 9 clusters

### 4.2 Evaluation as clustering

While the scores above can be indicative of the potential of different representations, they do not provide information on other aspects as cluster stability, purity, the amount of post-processing needed. Above all, in a completely unsupervised setting, the number of classes in

---

[1]We observed that these settings are sensitive to the chaining effect and there is probably room for improvement by experimenting with different task-specific clustering parameters.

the standard is not known and cluster quality (precision) plays an important role with respect to interpretability: it is easier unify two homogeneous clusters than to separate a noisy one. We ran complementary experiments with different numbers of clusters. Table 3 indicates results for 20 and 30 clusters. The input-output combination method still has an advantage, and concatenation and multiplication also perform well. However, the advantages over the baseline are less significant than when the number of clusters was identical to the standard.

| INPUT | #clust | P | R | F |
|---|---|---|---|---|
| $a_1$(base) | 20 | **0.3429** | 0.1642 | 0.2221 |
| add | 20 | 0.2434 | 0.1843 | 0.2098 |
| conc | 20 | 0.2718 | **0.2116** | 0.2380 |
| in-out.add | 20 | 0.2947 | 0.2076 | **0.2436** |
| mult | 20 | 0.3405 | 0.1886 | 0.2428 |
| in-out.mult | 20 | 0.2711 | 0.1432 | 0.1874 |
| diff | 20 | 0.2997 | 0.1161 | 0.1674 |
| $a_1$(base) | 30 | 0.3855 | 0.1712 | 0.2371 |
| add | 30 | 0.2714 | 0.1726 | 0.2110 |
| conc | 30 | 0.3331 | 0.1862 | 0.2389 |
| in-out.add | 30 | 0.3548 | 0.1947 | **0.2514** |
| mult | 30 | 0.3037 | **0.1995** | 0.2408 |
| in-out.mult | 30 | **0.3916** | 0.1304 | 0.1957 |
| diff | 30 | 0.3770 | 0.1318 | 0.1953 |

Table 3: Class-based results for 20 and 30 clusters

In the next runs, we measure how stable the different clustering solutions are with settings that are structurally very different from the standard, i.e. have significantly more clusters. Class-based precision and recall are less relevant measures in this setting, since they take the average over the nine standard classes and not over the produced clusters. We therefore decided to use *modified purity* (Korhonen et al., 2008), adapted for structurally different clustering solution. Modified purity gives the proportion of word couples belonging to the majority class $c$ in their cluster $k$:

$$PUR = \frac{\sum_{i=1}^{|K|} \max_j |w\ in\ k_i \cap w\ in\ c_j|}{\sum_{i=1}^{|K|} w\ in\ k_i} \quad (13)$$

Modified purity is indicative of the quality and interpretability of the clusters. It favorizes small clusters, but singleton clusters were discarded. This measure corresponds to predic-

tion accuracy in classification if we assign the majority label to clusters.

Two series of runs were evaluated: for 10, 20... up to 50, and for 60, 70... up to 100 clusters. Average results are reported. These scores indicate the average purity of clusters over different runs.

| INPUT | PUR |
|---|---|
| $a_1$(baseline) | 0.2940 |
| add | 0.3059 |
| conc | 0.3107 |
| in-out.add | **0.3235** |
| mult | 0.2575 |
| in-out.mult | 0.2119 |
| diff | 0.2297 |

Table 4: Cluster-based results, 10-50 clusters

| INPUT | PUR |
|---|---|
| $a_1$(baseline) | 0.3291 |
| add | 0.3578 |
| conc | **0.3737** |
| in-out.add | 0.3674 |
| mult | 0.3235 |
| in-out.mult | 0.2587 |
| diff | 0.3058 |

Table 5: Cluster-based results, 60-100 clusters

## 5 Discussion

The additive input-output combination shows promising results, especially when it comes to capturing the structure: in the clustering setting with 9 clusters, it identifies 8 classes out of 9 in the standard. This indicates a good potential in differentiating between relation types, especially because the standard is conceived in a way that it contains strongly related classes. It outperforms every other measure until the number of clusters grows significantly above those in the standard (Table 5), when the concatenative measure catches up. The baseline performs well, but additive methods all beat it, while difference is especially weak. Pairwise multiplication is good at recognizing the structure (7 classes out of 9), but not good at assigning elements.

Multiplicative methods show a fluctuating performance, especially the multiplicative input-output combination. This is due to the higher variance in similarities obtained by multiplica-

tion (in the case of input-output combination, 6 operands are multiplied), combined with the agglomerative clustering, which is sensitive to chaining.

The very high precision of the baseline method with a large number of clusters (Table 3) is noteworthy but not unexpected. Individual similarities have a strong precision for the easily identifiable clusters, while additional relational information is mostly expected to improve recall.

## 6 Conclusion and Future Work

We presented an experiment to identify relational similarities in word embedding compositions at a large scale, using an unsupervised approach. On the one hand, our results confirm the recent finding that many of the success attributed to vector arithmetics for analogies come from similarities between individual elements. On the other hand, taking second order similarity into account, we can improve relational similarities and take a step toward a meaningful representation for entity couples in a semantic relation.

The baseline performs well and is difficult to enrich with relation-aware information. The results indicate that the vector offset method for analogies, which replaces the pairwise similarity, is the least efficient in capturing generic semantic relations at a large scale. The vector difference representation does not conserve pairwise similarities and the offsets do not prove to be constant enough for unsupervised clustering. Multiplicative methods do not scale up either, although to a lesser extent: they capture some of the relational information, but this happens at the expense of losing precision from individual similarities. Pairwise similarities can be better exploited in an additive or concatenative setting. Moreover, they can be meaningfully complemented by including second order similarities without losing too much information for precise classification. The input-output combination measure coherently outperformed the other combinations in almost every setting, indicating a better potential for unsupervised experiments.

Unsupervised relation classification is a very challenging task for several reasons. Some relation instances are lexical by nature and, there-fore, can be expected to show up in the same cluster based on distributional cues. On the other hand, contextual relation instances tend to have relation-specific indicators when they co-occur, but their individual vectors will not reveal this information (unless they co-occur very often). Moreover, semantic relations differ with respect to the semantic constraints they put on their arguments. For instance, the second argument of the Content-Container relation tend to belong to a specific semantic class in the standard (*bag, box, trunk, case, drawer...*), while both arguments of the Cause-Effect relation are much freer (*gas, prices, pain, acts, species* and *pyrolysis, collapse, contraction, society, noise*). Any future development towards an automated unsupervised classification needs to take these aspects into account and work towards a hybrid solution by separating relations with semantically constrained arguments from free ones, as well as adapting the clustering method to handle outliers.

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
