# Peer review of "Exploring Vector Spaces for Semantic Relations"

_ACL 2017 — decision unknown_

[Official Review · Reviewer 1 · rating 4 · confidence 2]
soundness 3 · originality 3 · clarity 5 · impact 4 · substance 3 · appropriateness 5 · meaningful comparison 4 · presentation format Oral Presentation

- Strengths: The idea to investigate the types of relations between lexical
items is very interesting and challenging. The authors make a good argument why
going beyond analogy testing makes sense.  

- Weaknesses: The paper does not justify or otherwise contextualize the choice
of clustering for evaluation, rather than using a classification task, despite
the fact that classification tasks are more straightforward to evaluate. No
attempt is being made to explain the overall level of the results. How well
would humans do on this task (given only the words, no context)?

- General Discussion:

I have read the authors' response.

[Official Review · Reviewer 2 · rating 2 · confidence 4]
soundness 3 · originality 3 · clarity 3 · impact 4 · substance 2 · appropriateness 5 · meaningful comparison 4 · presentation format Poster

This paper investigates the application of distributional vectors of meaning in
tasks that involve the identification of semantic relations, similar to the
analogical reasoning task of Mikolov et al. (2013): Given an expression of the
form “X is for France what London is for the UK”, X can be approximated by
the simple vector arithmetic operation London-UK+France. The authors argue that
this simple method can only capture very specific forms of analogies, and they
present a measure that aims at identifying a wider range of relations in a more
effective way.

I admit I find the idea of a single vector space model being able to capture a
number of semantic relationships and analogies rather radical and infeasible.
As the authors mention in the paper, a number of studies already suggest for
the opposite. The reason is quite simple: behind all these models lies (some
form of) the distributional hypothesis (words in similar contexts have similar
meanings), and this poses certain limitations in their expressive abilities;
for example, words like “big” and “small” will always be considered as
semantically similar from a vector perspective (although they express opposite
meanings), since they occur in similar contexts. So I cannot see how the
example given in Figure 1 is relevant to the very nature of vector spaces (or
to any other semantic model for that matter!): there is a certain analogy
between “man-king”, and “woman-queen”, but asking from a word space to
capture “has-a” relationships of the form “owl-has-claws”, hence
“hospital-has-walls”, doesn’t make much sense to me.

The motivation behind the main proposal of the paper (a similarity measure that
involves a form of cross-comparison between vectors of words and vectors
representing the contexts of the words) is not clearly explained. Further, the
measure is tested on the relation categories of the SemEval 2010 task with
rather unsatisfactory results; in almost all cases, a simple baseline that
takes into account only partial similarities between the tested word pairs
present very high performance, with a difference from the best-performing model
which seems to me statistically insignificant. So from both a methodological
and an experimental perspective, the paper has weaknesses, and in its current
form seems to describe work in progress;  as such I am inclined against its
presentation in ACL.

(Formatting issue: The authors use the LaTeX styles for ACL 2016 — this
should be fixed in case the paper is accepted).

AUTHORS RESPONSE
================
Thank you for the clarifications. I am still not comfortable with the idea of a
metric or a vector space that tries to capture both semantic and relational
similarity, and I think you don't present enough experimental evidence that
your method works. I have to agree with one of the other reviewers that a more
appropriate format for this work would be a short paper.

[Official Review · Reviewer 3 · rating 2 · confidence 4]
soundness 3 · originality 3 · clarity 4 · impact 4 · substance 1 · appropriateness 5 · meaningful comparison 4 · presentation format Poster

This paper presents a comparison of several vector combination techniques on
the task of relation classification.

- Strengths:

The paper is clearly written and easy to understand.

- Weaknesses:

My main complaint about the paper is the significance of its contributions. I
believe it might be suitable as a short paper, but certainly not a full-length
paper.

Unfortunately, there is little original thought and no significantly strong
experimental results to back it up. The only contribution of this paper is an
'in-out' similarity metric, which is itself adapted from previous work. The
results seem to be sensitive to the choice of clusters and only majorly
outperforms a very naive baseline when the number of clusters is set to the
exact value in the data beforehand.

I think that relation classification or clustering from semantic vector space
models is a very interesting and challenging problem. This work might be useful
as an experimental nugget for future reference on vector combination and
comparison techniques, as a short paper. Unfortunately, it does not have the
substance to merit a full-length paper.